# Environmentally stable interface of layered oxide cathodes for sodium-ion batteries

Shaohua Guo[1,2], Qi Li[2], Pan Liu [3,4], Mingwei Chen[3,4] & Haoshen Zhou[1,2]

Sodium-ion batteries are strategically pivotal to achieving large-scale energy storage. Layered oxides, especially manganese-based oxides, are the most popular cathodes due to their high reversible capacity and use of earth-abundant elements. However, less noticed is the fact that the interface of layered cathodes always suffers from atmospheric and electrochemical corrosion, leading to severely diminished electrochemical properties. Herein, we demonstrate an environmentally stable interface via the superficial concentration of titanium, which not only overcomes the above limitations, but also presents unique surface chemical/electrochemical properties. The results show that the atomic-scale interface is composed of spinel-like titanium (III) oxides, enhancing the structural/electrochemical stability and electronic/ionic conductivity. Consequently, the interface-engineered electrode shows excellent cycling performance among all layered manganese-based cathodes, as well as high-energy density. Our findings highlight the significance of a stable interface and, moreover, open opportunities for the design of well-tailored cathode materials for sodium storage.

[1] Center of Energy Storage Materials & Technology, College of Engineering and Applied Sciences, National Laboratory of Solid State Microstructures, Collaborative Innovation Center of Advanced Microstructures, Nanjing University, Nanjing 210093, China. [2] Energy Technology Research Institute, National Institute of Advanced Industrial Science and Technology (AIST), Tsukuba 305-8568, Japan. [3] State Key Laboratory of Metal Matrix Composites, School of Materials Science and Engineering, Shanghai Jiao Tong University, Shanghai 200240, China. [4] WPI Advanced Institute for Materials Research, Tohoku University, Sendai 980-8577, Japan. Correspondence and requests for materials should be addressed to P.L. (email: panliu@sjtu.edu.cn) or to H.Z. (email: hszhou@nju.edu.cn)

Large-scale energy storage systems (ESS) are key to the smooth integration the renewable resources, such as wind and solar, into the grid. Among them, the electrochemical approach is regarded as a smart choice, efficiently advancing the grid reliability and utilization[1,2]. The application cost, service life, and efficiency should be the primary focus for stationary batteries, in contrast to power batteries. Over the past decades, lithium-ion batteries (LIBs) have explosively advanced, resulting in a shortage and prohibitively expensive cost of lithium resources. In contrast, sodium is one of the most abundant elements on earth. The abundant reserves and insertion mechanism of sodium make sodium-ion batteries (SIBs) an ideal alternative to LIBs in large-scale applications[3–10].

Recently, many efforts have been made to develop cathode materials for SIBs, such as layered oxides, polyanion compounds, and Prussian-blue analogs[11–18]. Among the various cathode materials, layered oxides of $Na_xTMO_2$ ($x \leq 1$, TM = Mn, Ni et al.) based on abundant materials have been mostly studied as cathodes for SIBs[19,20], and in particular, Mn-based materials meet requirements for low-cost stationary batteries without sacrificing energy density or safety[12,21–25]. Their low cost and high performance makes layered manganese-based oxides very promising cathode candidates for SIBs, which are also potentially competitive with $LiCoO_2$, i.e., the system most widely used in LIBs[26]. Some major problems exist in layered manganese-based oxides: (1) Water or carbon dioxide can be easily inserted into the interlayer sites due to the fragile interface when layered Na-containing oxides are exposed to air[27,28]. (2) The electro-chemically active Mn(III) associated with the disproportionation reaction ($2Mn^{3+} \rightarrow Mn^{2+} + Mn^{4+}$) causes the possible dissolution of Mn from cathode/electrolyte interface[29,30]. (3) The two abovementioned phenomena result in severe degradation of the layered structures and rapid fading of the reversible capacity.

Herein, we demonstrate the unique surface chemistry via Ti-enrichment-induced surface reconstruction in $NaMnTi_{0.1}Ni_{0.1}O_2$ (NMTN). The Ti(III)-concentrated spinel-like overlayer with atomic-scale thickness exhibits a distinct crystal and electronic structure, giving rise not only to higher electron and ion conductivity, but also to the chemical/electrochemical/thermal stabilization of the layered bulk materials. By contrast, layered $NaMnO_2$ (NM) shows a typical phase transition (O'3→Birnes-site) when exposed to air and increased impedance with cycling in cells. Consequently, the NMTN electrode demonstrates a high reversible capacity, superior rate capability, and outstanding cyclability, suggesting that the present material is a very promising candidate for cathode materials of SIBs. This facile strategy will contribute to the development of room-temperature SIB technology to achieve high-energy and high-power density.

## Results

**Bulk structures of NM and NMTN.** NM and NMTN samples were synthesized through typical solid-state reactions. The bulk structures of the two samples were characterized by scanning electron microscopy (SEM), X-ray diffraction (XRD), selected area electron diffraction (SAED), and electron energy-loss spectroscopy (EELS). NM samples show irregular particles (Supplementary Fig. 1a), a monoclinic O'3 structure (Supplementary Fig. 1b,c), and an even distribution of elemental Na and Mn (Supplementary Fig. 1d). The SEM image of the NMTN samples (Supplementary Fig. 2a) indicates secondary particles composed of layered sheets. Titanium/nickel doping causes the biphasic coexistence of hexagonal P2 and monoclinic O'3 in the bulk materials, according to XRD (Supplementary Fig. 2b) and SAED (Supplementary Fig. 2c) analyses. The refined XRD result is given out in Supplementary Table 1. In particular, the high-resolution elemental mapping using EELS (Supplementary Fig. 2d) demonstrates that Ti is significantly concentrated on the particle surface in comparison to the other elements, potentially tuning the unique surface chemistry.

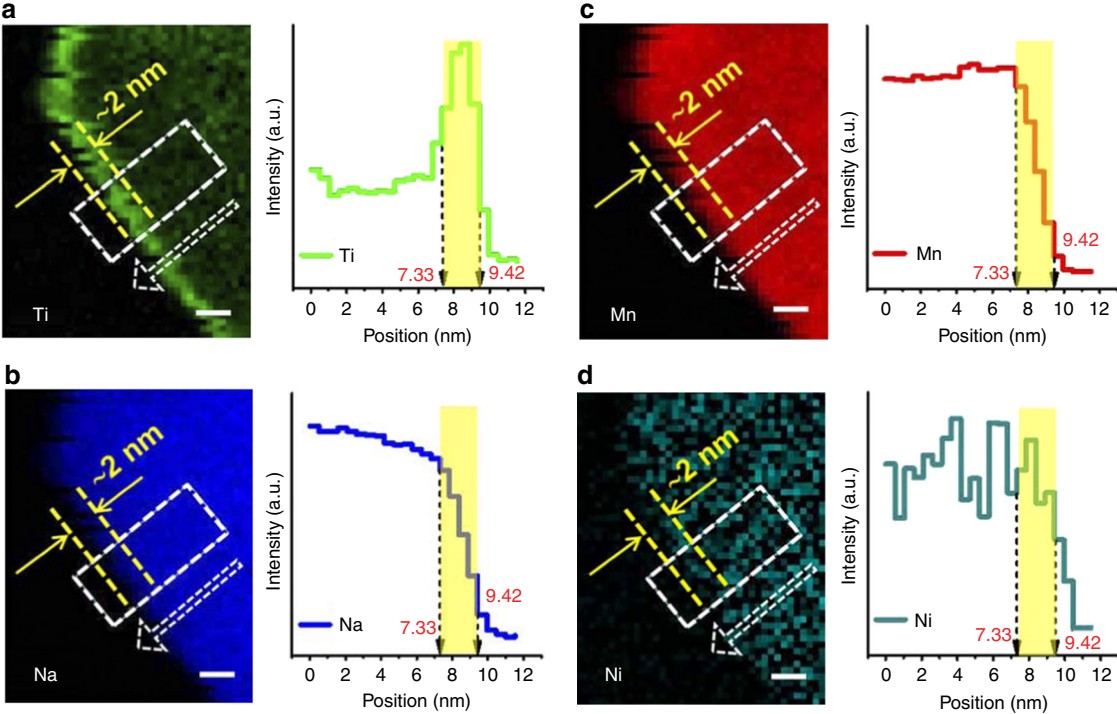

**Fig. 1** Titanium surface segregation. EELS chemical mapping of the NMTN samples and corresponding chemical compositions (*white rectangle*) for **a** Ti, **b** Na, **c** Mn, and **d** Ni elements. The *yellow dotted lines* and *vertical shadings* indicate the location of the interface. The results show that the interface 2 nm from the particle edge is enriched by titanium rather than other elements. Scale bar, 2 nm

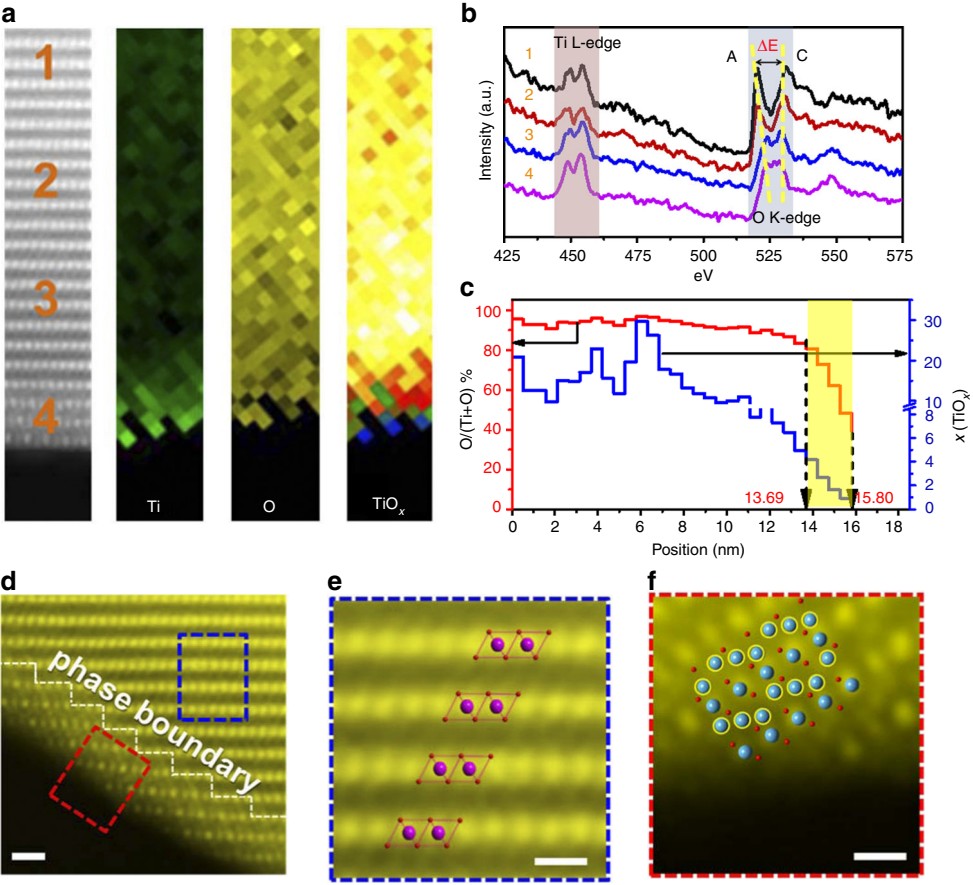

**Fig. 2** Oxidation state, composition, and crystalline phase of the local fine structure of the NMTN samples. **a** HADDF-STEM and EELS mapping images focusing on the Ti and O composition and chemical shift from the core to the shell for a typical location; the numerical marks indicate the selective spots for EELS. **b** Energy-loss near-edge spectrum (ELNES) of the Ti-L$_{2,3}$ (*vertical brown shading*) and O-K (*vertical blue shading*) edges, wherein the *yellow dotted lines* show peak A and peak C of the O-K edges. **c** Chemical composition of Ti and O, in which the *vertical yellow shading* represents the location of the interface; **d** STEM-HADDF image, indicative of the zigzag phase boundary; scale bar, 1 nm. **e** Magnified image extracted from the *blue rectangle* of **d**, assigned to the typical layered stacking in the bulk; scale bar, 0.5 nm. **f** Magnified image extracted from the *red rectangle* of **d**, showing the spinel-like stacking at the interface; scale bar, 0.5 nm

Surface segregation is common in oxide compounds and metallic alloys, and the driving force of segregation is the surface energy and strain energy in the bulk, respectively[31, 32]. A model calculation to determine surface segregation in oxide solid solutions was demonstrated by Kung in 1981[33], and it has been concluded that the substantial mismatch between the ionic radii of 'solvent' metal ions and 'solute' metal ions contributes to the surface segregation. In the simple statistical-mechanical mode of segregation[33], the total free energy F for a system consisting of several elements is written as:

$$F = \sum_i n_i^b g_i^b + n_i^s g_i^s - k_B T \ln \Omega \quad (1)$$

where $n_i^b$ and $n_i^s$ indicate the number of bulk and surface atoms of type $i$ with individual free energies $g_i^b$ and $g_i^s$, respectively. $\Omega$ represents the entropy due to mixing of the compounds. For a two-component system, an Arrhenius expression is obtained:

$$n_1^s / n_2^s = n_1 / n_2 \exp\left(-H_{seg}/k_B T\right) \quad (2)$$

where $H_{seg}$ is the heat of segregation. Equation 1 implies that surface segregation results from a competition between minimizing the total free energy by maximizing the entropy through evenly mixing several elements and minimizing the free energy of each element. From Equation 2, the enthalpy can be

determined as a function of the atomic fraction[32]. Mackrodt et al. also measured the heat of segregation of various binary oxide systems, and demonstrated a very large driving force toward the equilibrium segregation of calcium, namely, −50.7 kJ mol$^{-1}$ for the heat of Ca segregation on the surface of MgO[34]. Thus, if there is little significant segregation, the terminal layer is a polar surface, i.e., the surface is likely to be unstable toward surface reconstruction. These differences at the surface suggest that under equilibrium conditions, the surface will not have a composition and/or structure representative of the bulk and the 'solute' ions are inclined to segregate on the surface[32]. In NMTN samples, 'solute' titanium ions show significant surface segregation despite the mismatch between the nickel and manganese ions. This is similar to the previous report, in which co-doping Fe and Ga in a high-voltage LiMn$_{1.5}$Ni$_{0.5}$O$_4$ cathode led to the preferential segregation of Ga rather than Fe[35]. Further studies will be conducted to determine detailed reasons why only one of the doped metal ions showed significant segregation. It is well known that surface segregation has a profound impact on the electronic and chemical properties of as-prepared materials, and thus, surface segregation can be utilized to obtain desirable materials[36].

**STEM-EELS characterization of the NMTN interface.** To accurately evaluate the evolution of the chemical composition, NMTN samples were scanned from the center to the surface by

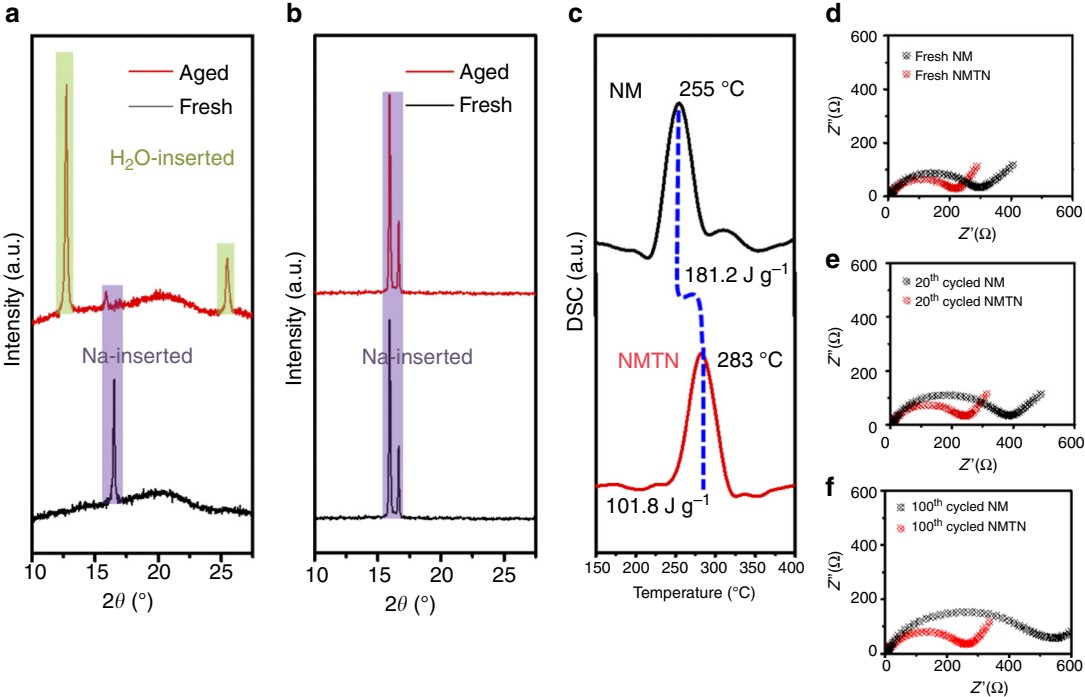

**Fig. 3** Chemical/electrochemical/thermal stability of the NM and NMTN samples. **a** Structural evolution of the NM samples exposed to humid air for 3 days. The results show the phase transition process (O'3-Birnessite) caused by $H_2O$ insertion; the *vertical purple* and *yellow-green shadings* indicate the Na-inserted and $H_2O$-inserted layered structures, respectively. **b** Structural evolution of the NMTN samples exposed to humid air for 3 days. The results indicate a stabilized layered-structure independent of air exposure. **c** DSC results of the desodiated NM and desodiated NMTN electrodes. **d**–**f** Impedance evolution of the NM and NMTN electrodes during cycling. The results show the severely diminished electrochemical properties for the NM electrode and negligible electrochemical evolution of the NMTN electrode

elemental-resolution EELS, and the corresponding concentration spectra were simultaneously collected. Figure 1a shows that Ti thinly disperses in the bulk, and instead remarkably gathers within 2 nm of the surface (marked in yellow), and the Ti content increases more than two-fold from the bulk to the surface. In contrast, Na, Mn, and Ni elements (Figs. 1b–d) are homogeneously distributed in the bulk, with no significant fraction on the overlayer. Therefore, we consider the interface especially the outermost shell, to be mostly composed of titanium oxide rather than other transition-metal oxides. Note that every particle investigated under the microscope shows a uniform and continuous titanium-concentrated interface without exception, indicative of the intrinsic surface Ti-enrichment in the NMTN samples. Such chemical nature allows the overall enclosure of bulk materials in contrast to general coating methods, as the latter always suffers from the partial exposure of bulk materials due to the discontinuities and inconsistencies in the coating.

**Atomic-scale characterization of the NMTN interface**. To understand the detailed electronic and crystal structure of the NMTN interface, we chose a typical location (Fig. 2a) for STEM and EELS characterization. The leftmost area of Fig. 2a shows four distinct regions from the bulk to the surface, which were identified by the energy-loss near-edge spectrum (ELNES) of the Ti-$L_{2,3}$ and O-K edges (Fig. 2b). It is evident that the two peaks of Ti $L_2$-$t_{2g}$ and Ti $L_3$-$t_{2g}$ are significantly suppressed at the interface, implying a valance change. The suppressed and broadened Ti $L_3$-$e_g$ peak also indicates the existence of highly distorted Ti–O octahedra. Estimated from the $t_{2g}/e_g$ ratio of the Ti-$L_2$ peak, the Ti valence diminishes and tends to evolve from a tetravalent to a trivalent state, indicative of a lower Ti oxidation state at the surface. Additionally, the O-K edge is relevant to the valence and lattice distortions, wherein $\Delta E$ represents the

distance between peak A and peak C. The suppression of peak A and decreased distance of $\Delta E$ also reflect a decrease in the Ti valence. Both the Ti-$L_{2,3}$ and O-K edges indicate the trivalent characteristics of surface-concentrated titanium, which yields enhanced electrical conductivity due to the presence of $d$-electrons[37]. The Mn-$L_{2,3}$ and Ni-$L_{2,3}$ peaks are consistent throughout the particles and are assigned to the $Mn^{3+}$ and $Ni^{2+}$ oxidation states (Supplementary Fig. 3). Moreover, X-ray photoelectron spectroscopy (XPS) analysis was also conducted to further confirm the valence state of the surface in NMTN samples and is shown in Supplementary Fig. 4. According to previous results, the Ti-$2p_{3/2}$ peaks at 456.6 and 458.3 eV simulated by a fitting process are attributed to $Ti^{3+}$ and $Ti^{4+}$, respectively;[38] a Mn-$2p_{3/2}$ peak is observed at 641.5 eV, which is representative of $Mn^{3+}$;[39] the Ni-$2p_{3/2}$ peak located at 855.6 eV confirmed the presence of $Ni^{2+}$[40]. Given that the XPS and ELNES techniques are relatively limited to the particle surface, these results mainly reflect the surface characteristics of the NMTN samples, and the core region, which possibly contains other valence states, such as tetravalent manganese, cannot be detected at present.

The stoichiometric ratio of oxygen and titanium was further estimated and is plotted in Fig. 2c. The proportion of oxygen in the core is relatively high, given the composition of the bulk material ($NaMn_{0.8}Ti_{0.1}Ni_{0.1}O_2$), and sharply decreases near the overlayer (marked in *yellow*), as observed in Fig. 1a. The composition of the outermost shell is determined to range from $TiO_{1.63}$ to $TiO_{0.66}$ (*yellow vertical shading* in Fig. 2c). The chemical composition of the titanium(III) oxide interface is clearly distinguished from that of the bulk and other titanium oxides, wherein the oxygen vacancies surrounded by titanium(III) ions can potentially yield good conductivity[41].

The crystal structure of the interface was identified via high-angle annular dark field scanning transmission electron

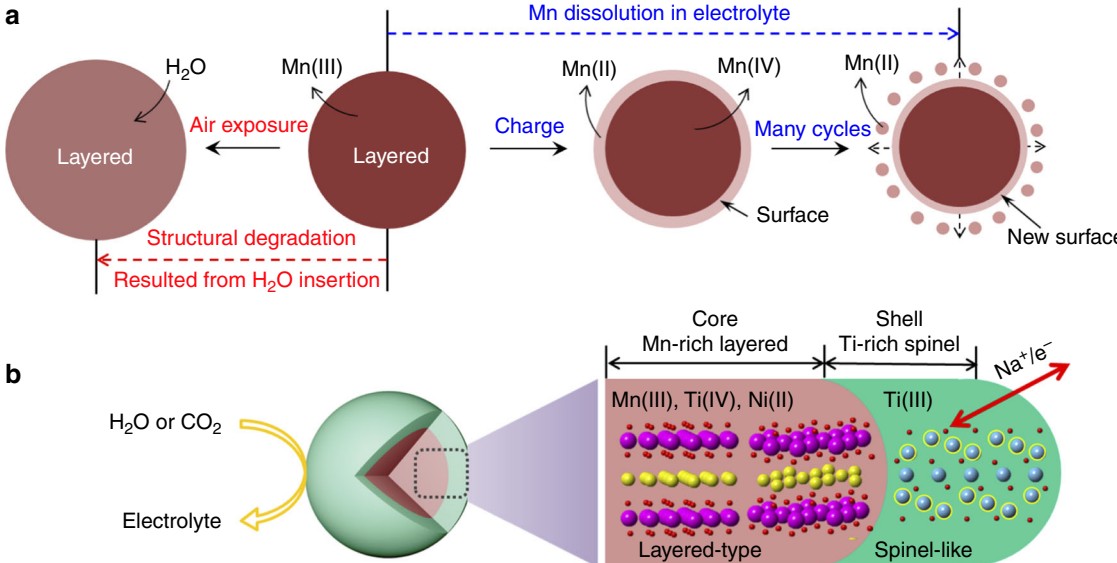

**Fig. 4** Diminished performance mechanism of the NM samples and protective mechanism of the NMTN samples. **a** Structural degradation of the NM samples subjected to attack from air (*red dotted left arrow*) and electrolyte (*blue dotted right arrow*). **b** 3D structural model of the NMTN samples composed of a Ti-rich spinel-like interface (*green shell*) and Mn-rich layered bulk (*brown core*). The layered bulk (*brown core*) is composed of P2 (*left*) and O′3 (*right*) phases. The Ti-rich spinel-like interface not only protects the bulk NMTN samples against atmospheric and electrochemical corrosion but also enhances the electrochemical properties via facile electron and ion transport

microscopy (HAADF-STEM). As the Z-contrast of the HAADF images is relevant to the atomic number, the bright dots in our study correspond to transition-metal ions, while the sodium and oxygen ions are nearly invisible. Figure 2d reveals the accurate atomic arrangements, wherein a zigzag phase boundary between two distinct stackings is visible. To distinguish the two stackings, the magnified images of Figs. 2e and f can be well assigned to the typical layered and spinel-like structures, respectively, corresponding to the inset atomic models. Moreover, the corresponding simulated HAADF-STEM images (Supplementary Fig. 5) and phase boundary atomic model (Supplementary Fig. 6) are given. The titanium sites in the normal spinel structure are fully occupied, and many of titanium sites are periodically partially occupied in our spinel-like titanium oxide layers, and thus act as Na-ion channels during the charge-discharge process. Note that we examined over 20 particles in the NMTN samples, and did not observe the coexistence of spinel/P2 phases; we speculate that the P2 phase resides in the inner core of the NMTN samples. After observation of the thinning area in the central part of the bulk, the biphasic structure of monoclinic O′3 and hexagonal P2 is found to exist in the inner core (Supplementary Fig. 7), confirming the structural model of spinel@O′3@P2 from the surface to the center. We also investigated the NM samples through STEM, and no distinct interface was detected (Supplementary Fig. 8). As discussed above, the Ti surface-concentration results in a well-refined titanium oxide layer with a unique electronic and crystal structure, wherein trivalent titanium is favorable and the improved electrical conductivity and spinel-like structure facilitates ion transport in the three-dimensional (3D) channel[42].

**Stability of the NM and NMTN Samples**. To investigate the chemical/electrochemical/thermal stability of the NM and NMTN samples, atmospheric aging experiments, Mn dissolution experiments, differential scanning calorimetry (DSC) and impedance experiments were performed. The NM and NMTN samples were exposed to humid air for 3 days, dried at 100 °C overnight and then analyzed by XRD. As shown in Fig. 3a, the

NM samples significantly changed. The main peak of the Na-inserted layered phase almost disappears and is accompanied by a newly formed peak at ~12°, which corresponds to the $H_2O$-inserted layered phase (Birnessite-type structure). The crystal structure is shown in Supplementary Fig. 9. The analysis also indicates that the interlayer spacing is mostly enlarged due to water molecule intercalation[27]. The NMTN samples (Fig. 3b) show the consistent diffraction pattern without the presence of new phases, demonstrating the protective role of titanium(III) oxide interface. The resistance to water intercalation may be due to the spinel-like structure that obstructs the intercalation of water.

We also investigated the dissolution of Mn in SIBs, which is a common concern for LIBs[29, 30]. As the dissolution–migration–deposition process continuously occurs in the electrode materials with cycling, the concentration of deposited Mn in the Na anodes was measured by the inductively coupled plasma (ICP) technique. Mn deposition is found in the cycled Na anode assembled with NM rather than NMTN (Supplementary Table 2). Additionally, Mn(II) species are detected at the surface of the cycled NM electrode due to the disproportionation of trivalent manganese (Supplementary Fig. 10), whereas the titanium oxide interface remains unchanged in the cycled NMTN electrode (Supplementary Fig. 11). Additionally, the conductivity is also enhanced due to the unique spinel-like titanium(III) oxide surface. Impedance spectra (Nyquist plots) of the NM and NMTN electrodes were obtained to evaluate the influence of Mn dissolution on the electrochemical properties. The equivalent circuit based on the Randles model and typical fitting pattern are shown in Supplementary Fig. 12. $R_e$ is the electrolyte resistance, and $R_f$ and $C_f$ are the resistance and capacitance of the surface film formed on the anodes, respectively. $R_{ct}$ and $C_{dl}$ are the charge-transfer resistance and double-layer capacitance, respectively. $Z_w$ is the Warburg impedance related the diffusion of sodium ions into the bulk electrodes[43]. As summarized in Supplementary Table 3, the $R_{ct}$ of the NMTN electrode (180.1 Ω) is smaller than that of the NM electrode (259.1 Ω), indicating that charge transfer on the surface of NMTN is faster than that on NM. The enhanced sodium

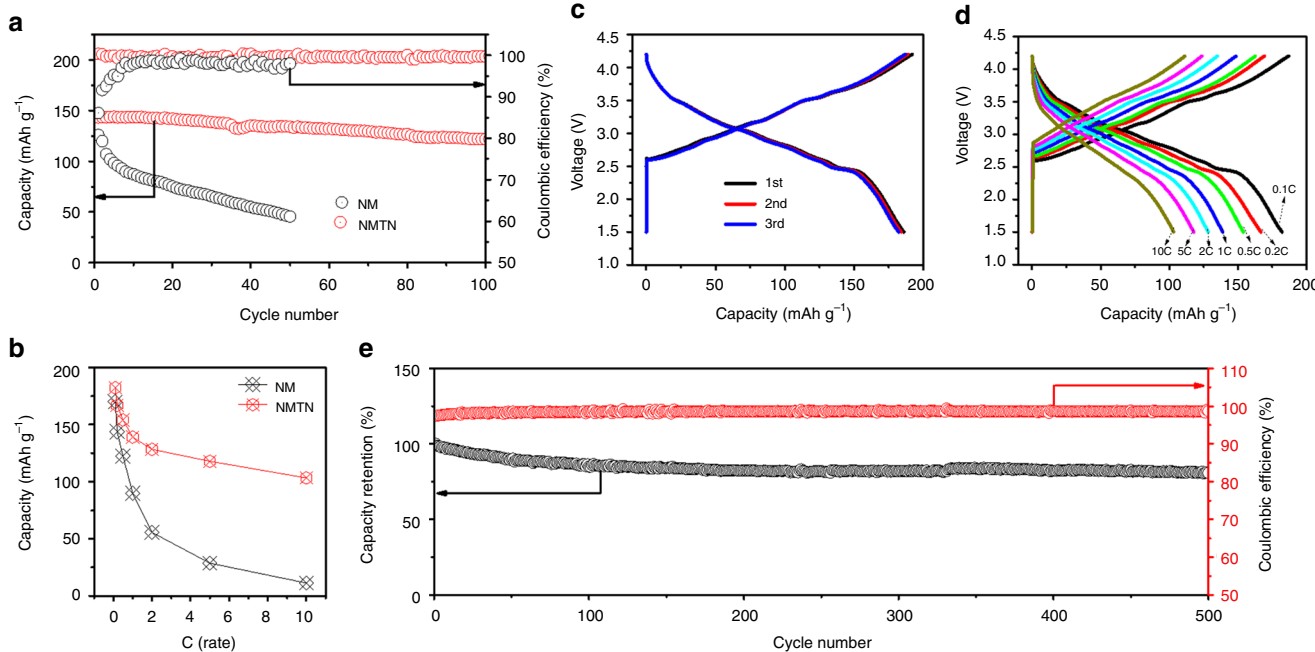

**Fig. 5** Electrochemical performance of the as-prepared NM and NMTN cathodes. **a** Cycling performance of the NMTN electrode at 0.5-C rate, with that of the NM electrode for reference; **b** Rate capability of the NMTN electrode with that of NM electrode for reference; **c** Typical potential profiles of NMTN at 0.1-C rate; **d** Potential profiles of NMTN at different rates. **e** The cycling performance and Coulombic efficiency of NMTN at 5-C rate

diffusion is due to the interface engineering of NMTN, wherein the spinel-like structure creates 3D channels for sodium migration[42]. Comparing Fig. 3e and f, a remarkable divergence in the change in impedance of the unmodified and surface-modified cathodes upon cycling is observed. The $R_{ct}$ of the NM electrode rapidly increases with cycling, and the $R_f$ of the anode also increases from 21.4 to 128.2 Ω after 100 cycles. The increase in $R_{ct}$ is caused by a structural change on the cathode surface due to Mn dissolution, and the increase in $R_f$ results from the compositional change of the solid electrolyte interface at the anode due to $Mn^{2+}$ deposition. In contrast, the increase in impedance was negligible for the surface-engineered NMTN electrode. There results clearly demonstrate that surface engineering is effective for suppressing the Mn dissolution–migration–deposition process, leading to a negligible increase in impedance for both the cathode and anode of NMTN-based cells.

Thermal stability is another important factor that should be considered prior to the use of electrode materials in alternative power sources. The NM and NMTN electrodes were first charged to 4.2 V (Fig. 3c), and then, the exothermic reactions were analyzed by DSC. The electrochemically desodiated NMTN electrode generates less heat at higher temperatures (101.8 J g$^{-1}$ at 283 °C) compared with the NM electrode (181.2 J g$^{-1}$ at 255 °C). Due to the stable spinel-like titanium oxide interface, the exothermic reaction of the NMTN samples appears at higher temperatures and releases less heat. In contrast, the side reaction between Mn species and the electrolyte may be detrimental to the thermal stability of the NM samples.

**Mechanism of the enhanced stability of the NMTN samples.** Based on the above results, we propose a mechanism for the structural evolution of the layered manganese oxides (NM samples) and Ti-protected oxides (NMTN samples) when exposed to air and cycled in electrolyte. As shown in Fig. 4a, the surface of the NM samples is very sensitive to environmental conditions such as humidity and electrolyte. Water in air can

easily insert into the NM samples, which enlarges the interlayer spacing and leads to the overall degradation of the Na-inserted phase. It has been reported that Birnessite containing water shows better cyclability than Birnessite in which the water was eliminated[44]; however, note that Birnessite Na$_x$MnO$_2 \cdot n$H$_2$O has a quite large layer-spacing ~7 Å and itself requires lattice water to be contained in the interlayer[45], substantially differing from typical Na$_x$MnO$_2$ characterized by O- or P-type structures. A reduction in sodium hosts due to potential side reactions between the electrolyte and extracted lattice water during cycling will severely deteriorate the battery performance. However, the soluble Mn(II) resulting from complex corrosion dissolves in electrolyte, resulting in the loss of active components, and then deposits on the Na anode, resulting in increased impedance, all of which are detrimental to battery performance. In contrast, the spinel-structured titanium-enriched interface could play a critical role in protecting against corrosion from air and electrolyte, thereby providing the chemical/electrochemical/thermal stability to the high-activity Mn(III) bulk (Fig. 4b). Note that the surface enrichment is essentially spontaneous, and spinel-structured titanium oxide could grow in situ around the entirety of the NMTN samples, effectively blocking Mn species from exposure to air/electrolyte. In addition to the titanium-segregated interface greatly suppressing electrochemical degradation, the simultaneous substitution of Ni also maintains the charge balance and high activity of the NMTN samples and reduces Mn$^{3+}$ distortion in the layered bulk, both of which jointly contribute to the improved electrochemistry. More importantly, the Ti(III)-concentrated spinel-like overlayer exhibits a unique electronic and crystal structure and gives rise to high ion and electron conductivity, which is very beneficial to the electrochemical performance. In contrast to general coating methods, which always generate higher resistance blocking ion and electron transport channels and diminishing the electrochemical properties, the doping-induced Ti-surface-concentration is economically

cost-saving and broadly applicable. It is worth noting that besides Ti surface segregation, the NMTN and NM cathodes have other differences, such as the synthetic temperature and chemical composition; however, these differences are nearly independent of the environmental corrosion occurring at the surface. Ti surface segregation provides the most protection against the surface side reactions and potentially contributes to the improved electrochemical performance.

**Cell performance of the NM and NMTN electrodes**. The electrochemical performances of the NM and NMTN cathodes were comparatively studied as shown in Fig. 5a, b. Figure 5a shows the cycling performance and Coulombic efficiency of the NM and NMTN electrodes at 0.5-C rate, respectively. A capacity retention of 36% is obtained for the NM electrode after 50 cycles, in stark contrast to that of 85% for the NMTN electrode after 100 cycles. The dramatically enhanced cyclic stability further proves the merits of titanium segregation at the surface and Ni substitution in the bulk of the NMTN samples. Comparative plots of the reversible capacity vs. C-rate between the NM and NMTN electrodes are given in Fig. 5b. At the same C-rate, all NMTN electrodes show higher discharge capacity than the NM electrodes. For instance, the NMTN and NM electrodes deliver a discharge capacity of 139 and 90 mAh g$^{-1}$ at 1-C rate, respectively. Even at a fast charging and discharging rate of 5 C (1000 mA g$^{-1}$), the NMTN electrode still shows a capacity of 118 mAh g$^{-1}$, which is four times higher than that of NM. The superior performance is attributed to the improved kinetics in the presence of the well-engineered interface, which facilitates fast electron transport to achieve timely Na-ion insertion/extraction.

Figure 5c–e illustrates the comprehensive sodium storage performance of the NMTN electrode. The typical potential profiles (1–3rd) are presented in Fig. 5c, wherein the almost overlapped curves imply the high reversibility of NMTN, with a large capacity of 186 mAh g$^{-1}$ at 0.1-C rate. Based on the average output voltage of ~3.1 V, the energy density of NMTN electrode is estimated to be approximately 576 Wh kg$^{-1}$, which is competitive with the state-of-the-art LIB cathodes. To further study the long-term cycling performance, NMTN electrode was evaluated for 500 cycles at 5-C rate (Fig. 5e). A capacity retention of 81% is obtained even after 500 Na-ion (de)insertion cycles, and the Coulombic efficiency over the whole-cycling process is ~98%, which is suitable for practical application. Consequently, the NMTN cathode with stable spinel@layered heterostructure has high-energy density, superior rate capability, and excellent cycling performance, demonstrating the great potential for large-scale application. A comparison is performed between the NMTN electrode and other layered phases[12, 23, 24, 40, 46–51] in Supplementary Table 4. It can be seen that the NMTN electrode shows the superior comprehensive electrochemical performance in terms of cycle life, reversible capacity, and rate capability.

The above experiments and analysis validate our design strategy, i.e., suppressing atmospheric and electrochemical corrosion via a well-engineered interface. In addition, this strategy is clearly more effective to achieve high-performance SIB electrodes as most layered Na-containing oxides suffer from the water insertion during aging in humid air[13]. This presents a striking contrast to LIBs, in which layered Li-containing oxides are very stable against water. Therefore, air-stable interfaces must be well constructed before the practical application of SIB electrodes[13]. Although the low cost and high activity of manganese adopted in SIBs provides further competitive advantages to LIBs, manganese dissolution, associated with the Jahn–Teller effect of trivalent manganese, severely deteriorates the comprehensive cell performance. The superficial

concentration of the doping element provides overall protection for the bulk materials, successfully separating the manganese species and corrosive electrolyte. Finally, the permeation of Na ions from the electrolyte to the electrode is regarded as a very slow process, and general coating treatments always further block ion and electron transport. The well-tailored spinel-like titanium (III) oxide interface supplies 3D channels for Na-ion diffusion and fast electron migration. These phenomena are based on the facile doping treatment approach. Doping-induced surface reconstruction provides new insight to tailor and modify electrode materials and should be further examined in future studies.

## Discussion

In summary, we demonstrate a precisely tailored Ti(III)-concentrated spinel-like interface in a layered manganese-based oxide that possesses numerous benefits. The all-titanium protecting layer effectively inhibits the contact between the Mn species and environmental conditions and suppresses the disproportionation of manganese. The Ti(III)-concentrated spinel-like overlayer with atomic-scale thickness enhances the electron and ion conductivity. Due to the protection of the entire surface, the chemical/electrochemical/thermal stability is remarkably enhanced. As a result, the Ti-surface-concentrated Mn-based electrode can supply a large capacity of 186 mAh g$^{-1}$, a rate capability of 118 mAh g$^{-1}$ (1000 mA g$^{-1}$, 5-C rate), and superior cycling stability, representing the best cycling performance for Mn-based oxide cathodes. Tailored cathode materials may lead to advanced SIBs that meet requirements for electric vehicles and renewable energy storage, as well as inspire the development of a wide range of different electrode materials.

## Methods

**Material preparation**. The NM and NMTN raw materials were synthesized by a traditional solid-state reaction. All solid chemical compounds (Wako Pure Chemicals Industries Ltd., Japan) were used without any purification. The starting materials of Na$_2$CO$_3$, Mn$_2$O$_3$, TiO$_2$, and NiO in stoichiometric proportion were well ground in an agate mortar at a rotation rate of 200 rpm for 20 h. Due to the volatility of sodium at high temperatures, an excess 5 wt% of Na$_2$CO$_3$ was added. The mixture was dried for 12 h at 100 °C. The obtained powders were pressed into pellets with a diameter of 16 mm, and then the pellets were heat-treated in oxygen flow. The NM samples were heated at 700 °C for 10 h, and the NMTN samples were heated at 900 °C for 15 h. The heated pellets were quenched to room temperature and then stored in an argon-filled glove box until use. The molar ratios of the metal ions in the NM and NMTN samples were chemically analyzed by ICP mass spectrometry (ICP-MS), and the Na/Mn ratio was 0.98:1.00 for NM samples, and the Na/Mn/Ti/Ni ratio was 0.95:0.80:0.11:0.10 for the NMTN samples, respectively, perfectly matching the ideal ratios of our designed materials. Given the mismatch between the sodium content and phases in the NMTN samples, there may be some transition-metal ion vacancies in the layered bulk materials[52].

**Characterizations**. The XRD patterns were measured using Cu Kα radiation on a Bruker D8 Advance Diffractometer (Germany), and refined using the GSAS + EXPGUI suite[53, 54]. SEM was performed to observe the particle size and morphology using a TOPCON DS-720 instrument (Japan). A 200 kV JEM-2100F electron microscope (JEOL) was equipped with two aberration correctors (CEOS GmbH) to achieve probe-forming and image-forming lens systems. Cs correctors were optimized for image observations and the point-to-point resolution of the STEM was 1.0 Å. Before the atomic resolution images were recorded, the lens aberrations were measured by evaluating the Zemlin tableau of an amorphous area. EELS measurements and elemental mappings were carried out using Gatan GIF Tridiem. The contrast in the HAADF-STEM images is basically proportional to the square of the atomic number and was acquired simultaneously using an annular-type STEM detector while ABF-STEM images were recorded by a STEM ABF detector simultaneously. The convergence angle was 25 mrad and the angular range of the collected electrons during HAADF imaging was ~60–250 mrad. XPS was recorded using a Thermo Fisher Scientific Model K-Alpha spectrometer equipped with Al Kα radiation (1486.6 eV). The cycled electrodes were scratched off from the electrodes of the sodium half-cells and subjected to ex situ XRD and ex situ STEM using the abovementioned equipment.

**STEM image simulation**. HAADF-STEM simulations were performed using the Win HREM software (HREM Research). The algorithm has been verified to be reliable for simulating HAADF-STEM images with a large atomic cell and for simulating Cs-corrected STEM images. In the calculations, the probe convergence angle was 25 mrad and the HAADF detector inner and outer angles were 100 and 267 mrad, respectively.

**Electrochemistry**. Electrochemical tests were carried out using CR2032 coin-type cells, consisting of the cathode and a sodium metal anode. The cathode electrodes were prepared with a weight ratio of 75% active material, 20% Teflonized acetylene black, and 5% polytetrafluoroethylene. Pellets for the half-cells were pressed into the form of aluminum screens ~2 mg in mass and 7 mm in diameter (the loading mass was ~1.3 mg cm$^{-2}$), which were then dried under vacuum at ~110 °C for 5 h before assembling the cells. The cells were assembled in a glove box filled with dry argon gas. The electrolyte was 1 mol dm$^{-3}$ NaClO$_4$ dissolved in propylene carbonate (Tomiyama Pure Chemical Industries, Japan) with 2 vol% fluorinated ethylene carbonate as an electrolyte additive. A glass fiber served as the separator of the sodium half-cells. Galvanostatic charge/discharge tests were performed using a Hokuto Denko HJ1001SD8 (Japan) battery tester, and all cells were operated at different current densities within a cut-off voltage window of 1.5–4.2 V vs. Na/Na$^+$ at 25 °C after a 12 h rest; 1 C corresponds to 200 mA g$^{-1}$ in the cell tests.

**Data availability**. The data supporting the findings of this study are available from the authors upon reasonable request.

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

## Acknowledgements

The financial support from NSF of China (21633003) is acknowledged. P.L. and M.C. are sponsored by JST-CREST "Phase Interface Science for Highly Efficient Energy Utilization", JST (Japan). We are thankful for discussions with Dr. Jiuhui Han, Dr. Guo-zhen Zhu, Mr. Kezhu Jiang, and Dr. Ping He.

## Author contributions

S.G. and H.Z. conceived the concept. S.G. and Q.L. performed the materials synthesis and electrochemical analysis. P.L. and M.C. conducted STEM and EELS characterization. S.G. analyzed the data and proposed the mechanism. S.G., P.L., and H.Z. wrote the paper. All authors discussed the results and commented on the manuscript.

## Additional information

**Competing interests:** The authors declare no competing financial interests.

