## [Peer Review File · Nature Communications]

Reviewers' comments:

Reviewer #1 (Remarks to the Author):

MS of "Environmentally-Stable Interface for Layered Oxide Cathodes of Sodium-Ion Batteries" described the surface structure of Na_xTMO_2 ($x \leq 1$, TM = Mn, Ni et al.). The authors found that the Ti (III) is homogeneously dispersed onto the Na_xTMO_2 when they prepared the material with a solid state method. Evidences from EELS, STEM, XRD, showed the the valence of Ti and Mn is +3. The thickness of Ti-rich surface layer is ca 20nm. The further electrochemical testing showed that the Mn dissolution and the sensitive of material to water and CO_2 were suppressed. The results is interesting. However, follows should be revised before MS being published.

The authors used a solid state method to prepare the materials, what is the machinist for Ti ion disperse homogeneously onto the the particle surface.

In figure 3, the plot of impedance spectra is not correct, normally, it length of z' and Z'' axis should be same since the shape of arc in the EIS is important for readers to understand the electrode processes. The equivalent circuit for simulation of impedance is not reasonable. The equivalent circuit should be based on Randles model which is widely used in explanation of EIS.

In figure 5 c and d. The label of y axis should be "potential", not voltage, since here authors discussed the cathode material, not the cell or battery.

Reviewer #2 (Remarks to the Author):

This Communication describes the study of a layered oxide $\text{NaMn}_{0.8}\text{Ti}_{0.1}\text{Ni}_{0.1}\text{O}_2$ comparing the results with those of the NaMnO_2 phase.

Comments:

- The literature is not updated. Some reviews in Na ion batteries are not mentioned. Considering the role developed by the Ti cations, other publications of layered oxides with Mn and Ti should be analyzed and referenced.
- Regarding the synthetic method used in the preparation of the samples it is difficult to accept that all Ti present in the sample is as Ti^{3+} . Have the authors observed the presence of impurities?
- Other surface techniques such as XPS should be used to corroborate the results.
- Why the existence of only Mn^{3+} is considered? Why the presence of Mn^{4+} is discarded?
- How the authors justify the presence of P2 phase with the amount of Na present in the sample?
- The mechanism described in Figure 4 is speculative. What is mean spinel phase?
- In some publications of layered oxides, it is described that the insertion of water increases the cyclability. A comment on this should be included.

In conclusion, the electrochemical behavior of this phase and basically the good cyclability it was expected. The authors have developed and interesting analysis of the surface but considering the role developed by the Ti, a comparative study with other Ti phases and P2 phases should be carried out before publication.

Reviewer #3 (Remarks to the Author):

In this manuscript, authors tried to achieve a high performance layered oxide positive electrode material for Na-ion batteries via the coating of spinel-like titanium oxides. Generally, Na-ion batteries are considered as one of candidates for large-scale energy storage systems because Na is much more abundant than lithium. Therefore, extensive attentions have been attracted to the research and

development of Na based electrode materials.

As one of the earliest and most characterized layered oxides, Na_xMnO_2 exhibits great cost benefit and potential electrochemical performance. However, the structure distortion that caused by Mn^{3+} Jahn-Teller effect and interstitial water is the main reason to deteriorate the performance of Na_xMnO_2 . Therefore, Guo et al. adopted Ni/Ti co-doping strategy to prepare the sample of $\text{NaMn}_{0.8}\text{Ti}_{0.1}\text{Ni}_{0.1}\text{O}_2$. Comparing with the layered NaMnO_2 , the as-designed material exhibits much better electrochemical performance and thermal stability. Furthermore, a Ti-concentrated spinel-like overlayer with atomic-scale thickness was observed on the surface of $\text{NaMn}_{0.8}\text{Ti}_{0.1}\text{Ni}_{0.1}\text{O}_2$ when authors characterized the sample with some advanced technologies, such as EELS and HADDF-STEM. It turns out that the sample of $\text{NaMn}_{0.8}\text{Ti}_{0.1}\text{Ni}_{0.1}\text{O}_2$ is a kind of Ni-doping and Ti_2O_3 coating layered oxide. Unfortunately, Ni doped Na_xMnO_2 or Ti substituted $\text{Na}_{2/3}\text{Ni}_{1/3}\text{Mn}_{2/3-x}\text{Ti}_x\text{O}_2$ have already been reported. The effects of Ni/Ti doping in this manuscript are almost as the same as the previous literatures. Hence, the originality and novelty of this work are limited. Indeed, Ti coating layer seems interest. However, in this manuscript, authors did not show any reason why Ti is significantly concentrated on the particle surface contrary to the other elements when $\text{NaMn}_{0.8}\text{Ti}_{0.1}\text{Ni}_{0.1}\text{O}_2$ was prepared by traditional solid-state reaction with the mixture of stoichiometric Na_2CO_3 , Mn_2O_3 , TiO_2 and NiO . Furthermore, Mn was partly replaced with Ni and Ti at the same time. Are all the benefits own to the Ti-concentrated interface or partially due to Ni doping? In the manuscript, the functions of Ni and Ti are not distinguished. The contribution of Ni doping is not mentioned. Therefore, it is difficult to say that this manuscript is well organized. Therefore, the referee does not recommend this manuscript to publish on Nature Communication.

Reviewers' comments:

Reviewer #1 (Remarks to the Author):

In authors have revised the MS carefully. The reviewer recommends MS to be published.

Reviewer #2 (Remarks to the Author):

The authors have answered the questions satisfactorily and the manuscript merits publication as it stands.

Reviewer #3 (Remarks to the Author):

The authors have answered the first and second comments satisfactorily. However, the referee does not agree with the reply for the third comment. Indeed, $\text{NaMn}_{0.8}\text{Ti}_{0.1}\text{Ni}_{0.1}$ exhibits outstanding electrochemical performance. However, the comparison between NaMnO_2 and $\text{NaMn}_{0.8}\text{Ti}_{0.1}\text{Ni}_{0.1}$ is not suitable. The differences between these two materials are multiple: i) Ti surface segregation, ii) synthesis temperature, iii) crystal structure and iv) chemical composition. In this case, it is quite difficult or even impossible to distinguish the contributions of these differences for the improvement of electrochemical performance. Authors added a sentence of "Ni simultaneous substitution also maintains the charge balance and the high activity of NMTN samples and reduces the Mn^{3+} distortion in the layered bulk" in the revised manuscript. However, only one sentence is insufficient to fully compare NaMnO_2 and $\text{NaMn}_{0.8}\text{Ti}_{0.1}\text{Ni}_{0.1}$. Therefore, referee judge that the present manuscript is not suitable for publication in Nature Communications.

REVIEWERS' COMMENTS:

Reviewer #3 (Remarks to the Author):

The revised one is acceptable for publication.

Referee 1

MS of “Environmentally-Stable Interface for Layered Oxide Cathodes of Sodium-Ion Batteries” described the surface structure of Na_xTMO_2 ($x \leq 1$, $\text{TM} = \text{Mn, Ni et al.}$). The authors found that the Ti (III) is homogeneously dispersed onto the Na_xTMO_2 when they prepared the material with a solid state method. Evidences from EELS, STEM, XRD, showed the valence of Ti and Mn is +3. The thickness of Ti-rich surface layer is ca 20nm. The further electrochemical testing showed that the Mn dissolution and the sensitive of material to water and CO_2 were suppressed. The results is interesting. However, follows should be revised before MS being published.

1 The authors used a solid state method to prepare the materials, what is the machinist for Ti ion disperse homogeneously onto the particle surface?

Reply:

Thanks for your valuable comments. Ti-element concentration onto the surface of particle is essentially elemental surface segregation, as we pointed out in the manuscript. Surface segregation is common in oxide compounds and metallic alloys, and has a profound impact on the electronic and chemical properties of these materials (Phys. Rev. B, 2000, 62, R14629). A model calculation on surface segregation for oxide solid solution has been demonstrated by Kung in 1981 (Surf. Sci., 1981, 110, 504-522). It has been proposed that the driving force of segregation is the surface energy and the strain energy in the bulk, respectively. In the simple statistical-mechanical mode of segregation, the total free energy F for a system consisting of several elements is written as:

$$F = \sum_i n_i^b g_i^b + n_i^s g_i^s - k_B T \ln \Omega \quad (1)$$

where n_i^b and n_i^s indicate the number of bulk and surface atoms of type i with individual free energies g_i^b and g_i^s , respectively. Ω represents the entropy due to the mixing of the compounds. For a two-component system, an Arrhenius expression is obtained:

$$n_1^s/n_2^s = n_1/n_2 \exp(-H_{seg}/k_B T) \quad (2)$$

Where H_{seg} is a heat of segregation. Equation 1 implies that surface segregation is a competition to minimize the total free energy by a maximization of entropy by mixing evenly several elements and a minimization of free energy of each element. From Eq. 2, the enthalpy can be determined as a function of the atomic fraction (Phys. Rev. B, 1999, 59, 13453). The significant titanium segregation for NMTN samples indicates that the surface is substantially different from the bulk in the free enthalpy.

Finally, it has been concluded that the substantial mismatch between the ionic radii of ‘solvent’ metal-ions and ‘solute’ metal-ions contributes to the surface segregation (Surf. Sci., 1981, 110, 504-522). Subsequently, Mackrodt et al. measured a heat of segregation for series of binary metal oxide system, further confirming the surface segregation by larger ‘solute’ metal-ions (J. Am. Ceram. Soc., 1989, 72, 1576-1583). A very large driving force toward the equilibrium segregation of calcium

is demonstrated, namely -50.7 kJ/mol for the heat of Ca segregation in surface of MgO. That means if there is little significant segregation, the terminal layer is a polar surface, i.e., a surface is likely to be unstable toward surface reconstruction. These differences at the surface suggest that, under equilibrium conditions, the surface will not have a composition and/or structure representative of the bulk, and the ‘solute’ ions are inclined to segregate in the surface (Acta Mater., 1997, 45, 5275-5284).

The view has been widely accepted by the following researchers when preparing In or Nb-doped TiO_2 (ACS Appl. Mater. Interfaces, 2012, 4, 6626–6634; Phys. Rev. B, 2000, 61, 13445), Ca, Sr, Pb-doped LaMnO_3 (Phys. Rev. B, 1999, 59, 13453; Phys. Rev. B, 2000, 62, R14629.), Cr, Fe, Ga-doped $\text{LiNi}_{0.5}\text{Mn}_{1.5}\text{O}_4$ (Electrochem. Commun., 2011, 13, 1213-1216; Chem. Mater., 2012, 24, 3720-3731) with surface segregation of doped ions. NMTN sample contains several transition-metal-ions such as manganese ‘solvent’ and titanium, nickel ‘solute’ without consideration of sodium segregation. Previous study has proven that it is a more complex system, wherein more than one ‘solute’ metal-ion replaces one ‘solvent’ metal-ion. Manthirm et al. found that when high-voltage cathode $\text{LiMn}_{1.5}\text{Ni}_{0.5}\text{O}_4$ is simultaneously substituted by Fe and Ga, only Ga-ions preferred to significantly segregate in the surface rather than Fe-ions, meanwhile Fe and Ga-ions both presented a remarkable segregation in case of separated doping by Fe or Ga (Chem. Mater., 2012, 24, 3720-3731). It is documented that when an oxide compound is simultaneously substituted by several other metal-ions, one of them would be finally segregated in the particle surface. Similar to that, titanium ions instead of nickel ions consequently showed a preferential segregation in the surface of NMTN samples. Detailed reasons are still unclear, and will be further investigated via the combination of experiments and simulations in the future.

The description is included accordingly in the revised manuscript:

Surface segregation is common in oxide compounds and metallic alloys, and the driving force of segregation is the surface energy and the strain energy in the bulk, respectively.^{31,32} A model calculation on surface segregation for oxide solid solution has been demonstrated by Kung in 1981,³³ and it has been concluded that the substantial mismatch between the ionic radii of ‘solvent’ metal-ions and ‘solute’ metal-ions contributes to the surface segregation. In the simple statistical-mechanical mode of segregation,³³ the total free energy F for a system consisting of several elements is written as:

$$F = \sum_i n_i^b g_i^b + n_i^s g_i^s - k_B T \ln \Omega \quad (1)$$

where n_i^b and n_i^s indicate the number of bulk and surface atoms of type i with individual free energies g_i^b and g_i^s , respectively. Ω represents the entropy due to the mixing of the compounds. For a two-component system, an Arrhenius expression is obtained:

$$n_1^s/n_2^s = n_1/n_2 \exp(-H_{seg}/k_B T) \quad (2)$$

where H_{seg} is a heat of segregation. Equation 1 implies that surface segregation is a competition to minimize the total free energy by a maximization of entropy by mixing evenly several elements and

a minimization of free energy of each element. From Eq. 2, the enthalpy can be determined as a function of the atomic fraction.³² Mackrodt et al. also measured a heat of segregation for various binary oxide systems, and a very large driving force toward the equilibrium segregation of calcium is demonstrated, namely $-50.7 \text{ kJ mol}^{-1}$ for the heat of Ca segregation in surface of MgO.³⁴ That means if there is little significant segregation, the terminal layer is a polar surface, i.e., a surface is likely to be unstable toward surface reconstruction. These differences at the surface suggest that, under equilibrium conditions, the surface will not have a composition and/or structure representative of the bulk, and the ‘solute’ ions are inclined to segregate in the surface.³² In NMTN samples, ‘solute’ titanium-ions show significant surface segregation despite of the presence of the mismatch between nickel and manganese-ions. This is similar to the previous report, where in case of bidoping Fe and Ga in high-voltage cathode $\text{LiMn}_{1.5}\text{Ni}_{0.5}\text{O}_4$, a preferential segregation of Ga rather than Fe was observed.³⁵ Detailed reasons why only one of doped metal-ions shows a significant segregation will be further investigated in the future. It is well known that surface segregation has a profound impact on the electronic and chemical properties of as-prepared materials, which thus is utilized for desirable materials.³⁶

2 In figure 3, the plot of impedance spectra is not correct, normally, it length of Z' and Z'' axis should be same since the shape of arc in the EIS is important for readers to understand the electrode processes. The equivalent circuit for simulation of impedance is not reasonable. The equivalent circuit should be based on Randles model which is widely used in explanation of EIS.

Reply:

Thanks for your helpful comments. We have equalized the Z' and Z'' axis in EIS plots of Fig. 3d-f, and also performed the simulation of impedance via the updated equivalent circuit based on the suggested Randles model (Supplementary Fig. 11). The description is included accordingly in the revised manuscript:

Figure 3 | Chemical/electrochemical/thermal stability of NM and NMTN samples. (a) Structural evolution for NM samples when exposed to moisture air for 3d. Results manifest the phase transition process (O'3-Birnessite) caused by H₂O insertion; the vertical purple and yellow-green shadings indicate the Na-inserted and H₂O-inserted layered structures, respectively. (b) Structural evolution for NMTN samples in moisture air for 3d. Results indicate a stabilized layered-structure independent of air exposure; (c) DSC results of desodiated NM and desodiated NMTN electrodes; (d), (e) and (f) Impedance evolution for NM and NMTN electrodes with cycling. Results show the severely declined electrochemistry for NM electrode and negligible electrochemical evolution for NMTN electrode, respectively.

Supplementary Figure 11 | Impedance spectra and fitting pattern of fresh NM electrode. The inset shows Randles equivalent circuit for simulation.

Supplementary table 3 | Fitting results for impedance

Samples	R_f (Ω)	R_{ct} (Ω)
Fresh cell (NM)	21.4	259.1
20 th cycled cell (NM)	39.5	322.7
100 th cycle cell (NM)	128.3	411.8
Fresh cell (NMTN)	16.9	180.1
20 th cycled cell (NMTN)	20.2	204.5
100 th cycle cell (NMTN)	26.3	235.8

The equivalent circuit based on Randles model and typical fitting pattern are shown in Supplementary Fig. 12. R_e is the electrolyte resistance, and R_f and C_f are the resistance and capacitance of the surface film formed on the anodes, respectively. R_{ct} and C_{dl} are the charge-transfer resistance and double-layer capacitance, respectively. Z_w is the Warburg impedance related the diffusion of sodium ions into the bulk electrodes.⁴³ As summarized in Supplementary Table 3, R_{ct} for NMTN electrode (180.1 Ω) is smaller than that for NM electrode (259.1 Ω), indicating that charge transfer on the surface of NMTN is faster than that of NM.

The R_{ct} of NM electrode rapidly increases with cycling, and the R_f of the anode also does from 21.4 to 128.2 Ω after 100 cycles. The increase in R_{ct} is caused by a structural change of the cathode surface because of Mn dissolution, and the increase in R_f results from the compositional change of the SEI at the anode due to Mn^{2+} deposition. In contrast, the increase in impedance was negligible for the surface-engineered NMTN electrode.

3 In figure 5 c and d. The label of y axis should be “potential”, not voltage, since here authors discussed the cathode material, not the cell or battery.

Reply:

Thanks for your valuable comments. We have changed to “potential” for indication of the cathode materials in Fig. 5c,d of the revised manuscript. The description of ‘voltage profile’ has been accordingly modified to ‘potential profile’ in the revised manuscript.

Figure 5 | Electrochemical performances of as-prepared NM and NMTN cathodes. (a) Cycle performance of NMTN electrode at 0.5C-rate with NM electrode as reference; (b) Rate capability of NMTN electrode with NM electrode as reference; (c) The typical potential profiles of NMTN at a 0.1C rate; (d) The potential profiles of NMTN at different rates; (e) The cycling performance and coulombic efficiency of NMTN at 5C-rate.

Referee 2

This Communication describes the study of a layered oxide $\text{NaMn}_{0.8}\text{Ti}_{0.1}\text{Ni}_{0.1}\text{O}_2$ comparing the results with those of the NaMnO_2 phase.

1 The literature is not updated. Some reviews in Na ion batteries are not mentioned. Considering the role developed by the Ti cations, other publications of layered oxides with Mn and Ti should be analyzed and referenced.

Reply:

Thanks for your helpful comments. We have updated the references including most of reviews on sodium-ion batteries (Sci. Technol. Adv. Mater., 2014, 15, 043501; Adv. Energy Mater., 2016, 6, 1600943; Funct. Mater. Lett., 2013, 6, 1330001; J. Electrochem. Soc., 2015, 162, A2589-A2604; Adv. Mater., 2015, 27, 5343–5364; Adv. Energy Mater., 2012, 2, 710-721; Angew. Chem. Int. Ed., 2015, 54, 3431-3448; Energy Environ. Sci., 2013, 6, 2338-2360; Adv. Funct. Mater., 2013, 23, 947-958; Energy Environ. Sci., 2016, 9, 2978; Energy Environ. Sci., 2012, 5, 5884-5901; Acc. Chem. Res., 2016, 49, 231-240; Chem. Rev., 2014, 114, 11636-11682). We have also included the literatures about layered oxides with Mn and Ti (Inorg. Chem., 2012, 51, 6211–6220; Phys. Chem. Chem. Phys., 2013, 15, 3304-3312; Chem. Commun., 2014, 50, 3677-3680; J. Mater. Chem. A, 2015, 3, 23261–23267; Chem. Commun., 2014, 50, 3677-3680; J. Mater. Chem. A, 2014, 2, 17268-17271; Energy Environ. Sci., 2015, 8, 1237–1244; Angew. Chem. Int. Ed., 2015, 54, 5894–5899) in the revised manuscript.

2 Regarding the synthetic method used in the preparation of the samples it is difficult to accept that all Ti present in the sample is as Ti^{3+} . Have the authors observed the presence of impurities?

Reply:

Thanks for your valuable comments. It should be emphasized that not all Ti-ions are trivalent, as stated in the manuscript, only Ti-ions segregated in the particle surface are demonstrated by trivalence. The valence information has been well characterized by ELNES as well as XPS techniques. XPS results also indicate that titanium ions are determined as the combination of trivalence and tetravalence, consistent with ELNES test. The description has been accordingly included in the revised manuscript:

Supplementary Figure 4 | XPS spectra of Ti 2p, Mn 2p, and Ni 2p in NMTN samples.

Moreover, XPS experiment was also conducted to further confirm the valence information of the surface in NMTN samples in Supplementary Fig. 4. According to previous results, the Ti-2p_{3/2} peaks at 456.6 eV and 458.3 eV simulated by fitting process are attributed to Ti³⁺ and Ti⁴⁺, respectively;³⁸ Mn-2p_{3/2} peak is observed at 641.5 eV, consistent with that of Mn³⁺;³⁹ Ni-2p_{3/2} peak located at 855.6 eV confirmed the presence of Ni²⁺.⁴⁰ Given that the XPS and ELNES techniques are relatively limited at the particle surface, these results mainly reflect the surface information of NMTN samples, and the core region possibly containing other valence states such as tetravalent manganese cannot be detected at present.

The impurities are not presented in NMTN samples on basis of the XRD and SAED experiments. XRD patterns demonstrate the biphasic coexistence of hexagonal P2 and monoclinic O'3 in NMNT samples, which is further confirmed by SAED characterization (Supplementary Fig. 2).

3 Other surface techniques such as XPS should be used to corroborate the results.

Reply:

Thanks for your helpful comments. We have performed the XPS experiment to further corroborate the valence state of particle surface. The description has been accordingly included in the revised manuscript:

X-ray photoelectron spectroscopy (XPS) was recorded using a Thermo Fisher Scientific Model K-Alpha spectrometer equipped with Al K α radiation (1486.6 eV).

Supplementary Figure 4 | XPS spectra of Ti 2p, Mn 2p, and Ni 2p in NMTN samples.

Moreover, XPS experiment was also conducted to further confirm the valence information of the surface in NMTN samples in Supplementary Fig. 4. According to previous results, the Ti-2p_{3/2} peaks at 456.6 eV and 458.3 eV simulated by fitting process are attributed to Ti³⁺ and Ti⁴⁺, respectively;³⁸ Mn-2p_{3/2} peak is observed at 641.5 eV, consistent with that of Mn³⁺;³⁹ Ni-2p_{3/2} peak located at 855.6 eV confirmed the presence of Ni²⁺.⁴⁰ Given that the XPS and ELNES techniques are relatively limited at the particle surface, these results mainly reflect the surface information of NMTN samples, and the core region possibly containing other valence states such as tetravalent manganese cannot be detected at present.

4 Why the existence of only Mn³⁺ is considered? Why the presence of Mn⁴⁺ is discarded?

Reply:

Thanks for your constructive comments. The present valence information are based on the combination of XPS and ELNES, and it should be noted that those special measurements themselves have the detectable limitation, wherein XPS and ELNES both focus on the surface region of the particle. During this region, we only observe the existence of trivalent Mn-ions based on the present experimental results. Taking into account the special core@shell configuration of NMTN samples, i.e., spinel@O₃@P2 from surface to center (Fig. 4b), it can be said with certainty that only spinel and O₃ structures are detectable in the surface region. Therefore, the observation of only Mn³⁺ is rational, since monoclinic O₃ with Mn³⁺ distortion is mostly composed by the approximate 'NaMnO₂'. However, we fully understand your concern that tetravalent manganese is possibly presented in NMTN samples due to the undetectable P2 phase in the inner core of NMTN samples. Therefore, we added the description concerning the valence information in the revised manuscript:

Given that the XPS and ELNES techniques are relatively limited at the particle surface, these results

mainly reflect the surface information of NMTN samples, and the core region possibly containing other valence states such as tetravalent manganese cannot be detected at present.

5 How the authors justify the presence of P2 phase with the amount of Na present in the sample?

Reply:

Thanks for your valuable comments. It is generally considered that sodium contents are relative to the phase structures for layered sodium-containing oxides Na_xTMO_2 , i.e. O3 phase is sodium-sufficient (e.g. $x=1$) compared with P2-phase (e.g. $x=0.8$). According to the ICP-MS test, NMTN sample is chemically composed by 0.95Na:0.80Mn:0.11Ti:0.10Ni, namely the empirical formula should be $\text{Na}_{0.94}(\text{Mn}_{0.79}\text{Ti}_{0.11}\text{Ni}_{0.10})\text{O}_{2+\delta}$, which seems to be near to that of O3 phase although NMTN samples still include ~40% P2 phase. However, it is recently reported that possible TM vacancies exist in layered manganese oxide (Angew. Chem. Int. Ed. 2016, 55, 1–5), wherein the empirical formula $\text{Na}_{0.66}\text{MnO}_{2+\delta}$ vs. chemical formula $\text{Na}_{0.59}\text{Mn}_{0.90}\square_{0.10}\text{O}_2$ (\square represents vacancy) are demonstrated. Accordingly, NMTN samples could be accordingly expressed as the chemical formula $\text{Na}_{0.85}(\text{Mn}_{0.71}\text{Ti}_{0.10}\text{Ni}_{0.09}\square_{0.10})\text{O}_2$ in case of 10% TM vacancies for bulk layered materials. Therefore, the mismatch between sodium contents and phases may be due to the existence of TM vacancies. The description has been accordingly included in the revised manuscript:

Given that the mismatch between sodium contents and phases in NMTN samples, there may be some vacancies of transition metal ions in the layered bulk materials.¹

6 The mechanism described in Figure 4 is speculative. What is mean spinel phase?

Reply:

Thanks for your valuable comments. Surface segregation of titanium induces unique surface chemistry, essentially differing from the layered bulk. The crystal structure, valence state, and composition of surface film are accurately characterized by advanced STEM and ELNES techniques. Results indicate that segregated titanium in the surface presents trivalent characteristic, favorable of enhanced electrical conductivity due to the presence of *d*-electrons; the composition of the surface film is determined to range from $\text{TiO}_{1.63}$ to $\text{TiO}_{0.66}$; spinel-like phase with a zig-zag phase boundary is clearly observed, potentially protecting the sensitive manganese species. Subsequently, we designed series of experiments to reveal the role of spinel-like phase in stabilizing chemical/electrochemical/thermal properties for NMTN samples. Results demonstrate that bulk structure is well sustained during moisture air for three days, Mn dissolution accompanied by increased impedance is significantly inhibited during electrochemical cycling, and the exothermic reaction appears at higher temperature. The chemical/electrochemical/thermal stability is in stark contrast to that of NM samples without Ti-enriched surface. Ti-rich spinel phase has proven to be greatly beneficial to the protective NMTN samples when suffering from the attack of the moisture air

and electrolyte, therefore we proposed protective mechanism for NMTN samples with Ti-rich spinel-like surface.

Figure 4 | Declined mechanism for NM samples and protective mechanism for NMTN samples. (a) Structural degradation of NM samples subjected to the attack from air (red dotted left arrow) and electrolyte (blue dotted right arrow); (b) 3D structural model of NMTN samples that is composed by the Ti-rich spinel-like interface (green shell) and Mn-rich layered bulk (brown core), respectively. The layered bulk (brown core) is composed by two stackings of P2 (the left) and O'3 (the right). Ti-rich spinel-like interface not only protects the bulk NMTN samples against the atmospheric and electrochemical corrosion, but also enhances the electrochemistry via the facile electron and ion transport.

7 In some publications of layered oxides, it is described that the insertion of water increases the cyclability. A comment on this should be included.

Reply:

Thanks for your helpful comments. We has cited and reviewed the suggested reference (Chem. Mater., 2015, 27, 3721–3725). The description has been accordingly included in the revised manuscript:

Although it has been reported that water-containing Birnessite shows better cyclability than water-eliminating Birnessite,⁴⁴ however, note that Birnessite $\text{Na}_x\text{MnO}_2 \cdot n\text{H}_2\text{O}$ has a quite large layer-spacing above $\sim 7\text{\AA}$ and itself requires lattice water to be pilled in interlayer,⁴⁵ substantially differing from the typical Na_xMnO_2 characterized by O or P-type structures.

8 In conclusion, the electrochemical behavior of this phase and basically the good cyclability it was expected. The authors have developed and interesting analysis of the surface but considering the role

developed by the Ti, a comparative study with other Ti phases and P2 phases should be carried out before publication.

Reply:

Thanks for your valuable comments. A comparative study between NMTN and other Ti phases (O3-Na_{0.8}Ni_{0.4}Ti_{0.6}O₂, P2-Na_{2/3}Ni_{1/3}Mn_{1/2}Ti_{1/6}O₂, O3-NaNi_{0.4}Fe_{0.2}Mn_{0.2}Ti_{0.2}O₂, O3-Na[NiCoFeTi]_{1/4}O₂) P2 phases (P2-Na_{0.7}MnO₂, P2-Na_{2/3}Fe_{1/2}Mn_{1/2}O₂, P2/O3-Na_{0.66}Li_{0.18}Mn_{0.71}Ni_{0.21}Co_{0.08}O₂, P2-Na_{0.80}Li_{0.12}Ni_{0.22}Mn_{0.66}O₂) has been carried out. The description has been accordingly included in the revised manuscript:

Supplementary table 4 | A comparison of some layered cathode materials

Cathode	Cycle life	Capacity	Rate
O'3/P2-NMTN (this work)	500 cycles (81%)	~186 mAh g ⁻¹	10C (~114 mAh g ⁻¹)
O3-NaNi _{1/2} Mn _{1/2} O ₂ ref. 24	50 cycles (75%)	~125 mAh g ⁻¹	1C (~105 mAh g ⁻¹)
P2-Na _{2/3} Ni _{1/3} Mn _{2/3} O ₂ ref. 50	50 cycles (95%)	~88 mAh g ⁻¹	2C (~62 mAh g ⁻¹)
P2-Na _{2/3} Ni _{1/3} Mn _{1/2} Ti _{1/6} O ₂ ref. 51	20 cycles (80%)	~127 mAh g ⁻¹	2C (~90 mAh g ⁻¹)
O3-Na _{0.8} Ni _{0.4} Ti _{0.6} O ₂ ref. 40	250 cycles (75%)	~83 mAh g ⁻¹	1C (~63 mAh g ⁻¹)
O3-NaNi _{0.4} Fe _{0.2} Mn _{0.2} Ti _{0.2} O ₂ ref. 47	200 cycles (84%)	~145mAh g ⁻¹	2C (~75 mAh g ⁻¹)
O3-Na[NiCoFeTi] _{1/4} O ₂ ref. 48	400 cycles (75%)	~116 mAh g ⁻¹	5C (~102 mAh g ⁻¹)
P2-Na _{2/3} MnO ₂ ref. 23	25 cycles (77%)	~175 mAh g ⁻¹	—
P2-Na _{2/3} Fe _{1/2} Mn _{1/2} O ₂ ref. 12	30 cycles (82%)	~190 mAh g ⁻¹	4C (~65 mAh g ⁻¹)
P2/O3-Na _{0.66} Li _{0.18} Mn _{0.71} Ni _{0.21} Co _{0.08} O ₂ ref. 46	150 cycles (75%)	~200 mAh g ⁻¹	5C (~69 mAh g ⁻¹)
P2-Na _{0.80} Li _{0.12} Ni _{0.22} Mn _{0.66} O ₂ ref. 50	50 cycles (~91%)	~119 mAh g ⁻¹	5C (~71 mAh g ⁻¹)

A comparison has been done between NMTN electrode and other layered phases^{12,23,24,40,46-51} in Supplementary Table 4. It can be seen that NMTN electrode shows the superior comprehensive electrochemical performance in terms of cycle life, reversible capacity, and rate capability.

In this manuscript, authors tried to achieve a high performance layered oxide positive electrode material for Na-ion batteries via the coating of spinel-like titanium oxides. Generally, Na-ion batteries are considered as one of candidates for large-scale energy storage systems because Na is much more abundant than lithium. Therefore, extensive attentions have been attracted to the research and development of Na based electrode materials.

As one of the earliest and most characterized layered oxides, Na_xMnO_2 exhibits great cost benefit and potential electrochemical performance. However, the structure distortion that caused by Mn^{3+} -Jahn-Teller effect and interstitial water is the main reason to deteriorate the performance of Na_xMnO_2 . Therefore, Guo et al. adopted Ni/Ti co-doping strategy to prepare the sample of $\text{NaMn}_{0.8}\text{Ti}_{0.1}\text{Ni}_{0.1}\text{O}_2$. Comparing with the layered NaMnO_2 , the as-designed material exhibits much better electrochemical performance and thermal stability. Furthermore, a Ti-concentrated spinel-like overlayer with atomic-scale thickness was observed on the surface of $\text{NaMn}_{0.8}\text{Ti}_{0.1}\text{Ni}_{0.1}\text{O}_2$ when authors characterized the sample with some advanced technologies, such as EELS and HADDF-STEM. It turns out that the sample of $\text{NaMn}_{0.8}\text{Ti}_{0.1}\text{Ni}_{0.1}\text{O}_2$ is a kind of Ni-doping and Ti_2O_3 coating layered oxide. Unfortunately, Ni doped Na_xMnO_2 or Ti substituted $\text{Na}_{2/3}\text{Ni}_{1/3}\text{Mn}_{2/3-x}\text{Ti}_x\text{O}_2$ have already been reported. The effects of Ni/Ti doping in this manuscript are almost as the same as the previous literatures. Hence, the originality and novelty of this work are limited. Indeed, Ti coating layer seems interest. However, in this manuscript, authors did not show any reason why Ti is significantly concentrated on the particle surface contrary to the other elements when $\text{NaMn}_{0.8}\text{Ti}_{0.1}\text{Ni}_{0.1}\text{O}_2$ was prepared by traditional solid-state reaction with the mixture of stoichiometric Na_2CO_3 , Mn_2O_3 , TiO_2 and NiO. Furthermore, Mn was partly replaced with Ni and Ti at the same time. Are all the benefits own to the Ti-concentrated interface or partially due to Ni doping? In the manuscript, the functions of Ni and Ti are not distinguished. The contribution of Ni doping is not mentioned. Therefore, it is difficult to say that this manuscript is well organized. Therefore, the referee does not recommend this manuscript to publish on Nature Communication.

Question 1: “Unfortunately, Ni doped Na_xMnO_2 or Ti substituted $\text{Na}_{2/3}\text{Ni}_{1/3}\text{Mn}_{2/3-x}\text{Ti}_x\text{O}_2$ have already been reported. The effects of Ni/Ti doping in this manuscript are almost as the same as the previous literatures. Hence, the originality and novelty of this work are limited.”

Reply:

Thanks for your valuable comments. It is worth pointing out that doping and surface segregation are quite distinct in terms of implication and efforts. Doping indicates that dopants are evenly introduced into a pure compound for tunable electrical properties while segregation refers to the enrichment of atoms or ions at a microscopic region especially in the surface, namely surface segregation in a materials system. Doping generally not involved in the structural evolution, could change the bulk

physical or chemical properties. Surface segregation often gives rise to the change of crystal configuration, as previously reported and here presented, which is mainly utilized to tune the surface physics/chemistry. The brightest spot of this work is that titanium surface segregation reconstructs a new interface for NMTN samples with unique surface chemistry, which is independent of doping in the bulk. Nevertheless, the Ni doped Na_xMnO_2 or Ti substituted $\text{Na}_{2/3}\text{Ni}_{1/3}\text{Mn}_{2/3-x}\text{Ti}_x\text{O}_2$ mentioned by the referee is not concerned with the segregation, where most of manganese ions are tetravalent. The fundamental issues involving in interfacial sensitivity in moisture air and organic electrolyte for layered Na-containing manganese (III) oxides have not been involved and addressed, despite the fact that those of Li-based manganese oxides have been concerned by scientists. NMTN sample presented in this work has proven that Ti-segregated surface is of great significance against the moisture air and organic electrolyte. Additionally, we have carefully compared NMTN electrode with the reported results (Inorg. Chem., 2012, 51, 6211-6220; Phys. Chem. Chem. Phys., 2013, 15, 3304-3312; Chem. Commun., 2014, 50, 3677-3680), and found that they are substantially different in terms of motivation, design strategy, and electrochemical performance.

- 1) Motivation: Ti substitution in NMTN samples is aimed at the concerns of sensitive surfaces in layered manganese oxides, i.e., water or carbon dioxide can be easily inserted into the interlayer sites due to the fragile interface when layered Na-containing oxides are exposed to air; the electrochemically active Mn(III) associated with the disproportionation ($2\text{Mn}^{3+} \rightarrow \text{Mn}^{2+} + \text{Mn}^{4+}$) causes the possible dissolution of Mn from cathode/electrolyte interface. In contrast, Ni doped Na_xMnO_2 or Ti substituted $\text{Na}_{2/3}\text{Ni}_{1/3}\text{Mn}_{2/3-x}\text{Ti}_x\text{O}_2$ are designed to gain the better cycling performance.
- 2) Design strategy: In the reported materials O3- $\text{NaNi}_{1/2}\text{Mn}_{1/2}\text{O}_2$, P2- $\text{Na}_{2/3}\text{Ni}_{1/3}\text{Mn}_{2/3}\text{O}_2$ and $\text{Na}_{2/3}\text{Ni}_{1/3}\text{Mn}_{2/3-x}\text{Ti}_x\text{O}_2$ (Inorg. Chem., 2012, 51, 6211-6220; Phys. Chem. Chem. Phys., 2013, 15, 3304-3312; Chem. Commun., 2014, 50, 3677-3680), manganese ions mostly present trivalent, and the electrochemical reactions are based on the redox of $\text{Ni}^{2+/4+}$, independent of Mn^{3+} distortion and dissolution. In our work, a small amount of Mn^{3+} is simultaneously substituted by Ti^{4+} and Ni^{2+} , and Mn^{3+} still dominants in the bulk materials. A stable Ti-concentrated interface for NMTN samples is obtained due to surface segregation not only to address the structural declination and Mn dissolution of layered manganese oxides.
- 3) Electrochemical performance: Consequently, the interface-engineered electrode shows the best cycling performance among all layered manganese-based cathodes as well as performing high energy density. The electrochemical performance of NMTN electrode has been compared with Ni doped Na_xMnO_2 (O3- $\text{NaNi}_{1/2}\text{Mn}_{1/2}\text{O}_2$ and P2- $\text{Na}_{2/3}\text{Ni}_{1/3}\text{Mn}_{2/3}\text{O}_2$) and Ti substituted $\text{Na}_{2/3}\text{Ni}_{1/3}\text{Mn}_{1/2}\text{Ti}_{1/6}\text{O}_2$ (the best of $\text{Na}_{2/3}\text{Ni}_{1/3}\text{Mn}_{2/3-x}\text{Ti}_x\text{O}_2$ system), which is highlighted as follows. The manuscript has been modified to include the description:

Supplementary table 4 | A comparison of some layered cathode materials

Cathode	Cycle life	Capacity	Rate
O'3/P2-NMTN (this work)	500 cycles (81%)	~186 mAh g ⁻¹	10C (~114 mAh g ⁻¹)
O3-NaNi _{1/2} Mn _{1/2} O ₂ ref. 24	50 cycles (75%)	~125 mAh g ⁻¹	1C (~105 mAh g ⁻¹)
P2-Na _{2/3} Ni _{1/3} Mn _{2/3} O ₂ ref. 50	50 cycles (95%)	~88 mAh g ⁻¹	2C (~62 mAh g ⁻¹)
P2-Na _{2/3} Ni _{1/3} Mn _{1/2} Ti _{1/6} O ₂ ref. 51	20 cycles (80%)	~127 mAh g ⁻¹	2C (~90 mAh g ⁻¹)
O3-Na _{0.8} Ni _{0.4} Ti _{0.6} O ₂ ref. 40	250 cycles (75%)	~83 mAh g ⁻¹	1C (~63 mAh g ⁻¹)
O3-NaNi _{0.4} Fe _{0.2} Mn _{0.2} Ti _{0.2} O ₂ ref. 47	200 cycles (84%)	~145mAh g ⁻¹	2C (~75 mAh g ⁻¹)
O3-Na[NiCoFeTi] _{1/4} O ₂ ref. 48	400 cycles (75%)	~116 mAh g ⁻¹	5C (~102 mAh g ⁻¹)
P2-Na _{2/3} MnO ₂ ref. 23	25 cycles (77%)	~175 mAh g ⁻¹	—
P2-Na _{2/3} Fe _{1/2} Mn _{1/2} O ₂ ref. 12	30 cycles (82%)	~190 mAh g ⁻¹	4C (~65 mAh g ⁻¹)
P2/O3-Na _{0.66} Li _{0.18} Mn _{0.71} Ni _{0.21} Co _{0.08} O ₂ ref. 46	150 cycles (75%)	~200 mAh g ⁻¹	5C (~69 mAh g ⁻¹)
P2-Na _{0.80} Li _{0.12} Ni _{0.22} Mn _{0.66} O ₂ ref. 49	50 cycles (~91%)	~119 mAh g ⁻¹	5C (~71 mAh g ⁻¹)

A comparison has been done between NMTN electrode and other layered phases^{12,23,24,40,46-51} in Supplementary Table 4. It can be seen that NMTN electrode shows the superior comprehensive electrochemical performance in terms of cycle life, reversible capacity, and rate capability.

To summarize, the interface issues of layered manganese oxides always inflicting the atmospheric and electrochemical corrosion is for the first time proposed, and well addressed via delicate design of stable Ti-concentrated interface based on the surface segregation in this work. We believe that the result of Ti-surface-concentrated NMTN represents a great breakthrough through nanoscale interface engineering, which undoubtedly of significant originality and novelty.

Question 2: “Indeed, Ti coating layer seems interest. However, in this manuscript, authors did not show any reason why Ti is significantly concentrated on the particle surface contrary to the other elements when NaMn0.8Ti0.1Ni0.1O2 was prepared by traditional solid-state reaction with the mixture of stoichiometric Na2CO3, Mn2O3, TiO2 and NiO.”

Reply:

Thanks for your valuable comments. Ti-element concentration onto the surface of particle is essentially elemental surface segregation, as we pointed out in the manuscript. Surface segregation is common in oxide compounds and metallic alloys, and has a profound impact on the electronic and chemical properties of these materials (Phys. Rev. B, 2000, 62, R14629). A model calculation on surface segregation for oxide solid solution has been demonstrated by Kung in 1981 (Surf. Sci., 1981, 110, 504-522). It has been proposed that the driving force of segregation is the surface energy and the strain energy in the bulk, respectively. In the simple statistical-mechanical mode of segregation, the total free energy F for a system consisting of several elements is written as:

$$F = \sum_i n_i^b g_i^b + n_i^s g_i^s - k_B T \ln \Omega \quad (1)$$

where n_i^b and n_i^s indicate the number of bulk and surface atoms of type i with individual free energies g_i^b and g_i^s , respectively. Ω represents the entropy due to the mixing of the compounds. For a two-component system, an Arrhenius expression is obtained:

$$n_1^s/n_2^s = n_1/n_2 \exp(-H_{seg}/k_B T) \quad (2)$$

Where H_{seg} is a heat of segregation. Equation 1 implies that surface segregation is a competition to minimize the total free energy by a maximization of entropy by mixing evenly several elements and a minimization of free energy of each element. From Eq. 2, the enthalpy can be determined as a function of the atomic fraction (Phys. Rev. B, 1999, 59, 13453). The significant titanium segregation for NMTN samples indicates that the surface is substantially different from the bulk in the free enthalpy.

Finally, it has been concluded that the substantial mismatch between the ionic radii of ‘solvent’ metal-ions and ‘solute’ metal-ions contributes to the surface segregation (Surf. Sci., 1981, 110, 504-522). Subsequently, Mackrodt et al. measured a heat of segregation for series of binary metal oxide system, further confirming the surface segregation by larger ‘solute’ metal-ions (J. Am. Ceram. Soc., 1989, 72, 1576-1583). A very large driving force toward the equilibrium segregation of calcium is demonstrated, namely -50.7 kJ/mol for the heat of Ca segregation in surface of MgO. That means if there is little significant segregation, the terminal layer is a polar surface, i.e., a surface is likely to be unstable toward surface reconstruction. These differences at the surface suggest that, under equilibrium conditions, the surface will not have a composition and/or structure representative of the bulk, and the ‘solute’ ions are inclined to segregate in the surface (Acta Mater., 1997, 45, 5275-5284).

The view has been widely accepted by the following researchers when preparing In or Nb-doped TiO₂ (ACS Appl. Mater. Interfaces, 2012, 4, 6626–6634; Phys. Rev. B, 2000, 61, 13445), Ca, Sr, Pb-doped LaMnO₃ (Phys. Rev. B, 1999, 59, 13453; Phys. Rev. B, 2000, 62, R14629.), Cr, Fe, Ga-doped LiNi_{0.5}Mn_{1.5}O₄ (Electrochem. Commun., 2011, 13, 1213-1216; Chem. Mater., 2012, 24, 3720-3731) with surface segregation of doped ions. NMTN sample contains several transition-metal-ions such as manganese ‘solvent’ and titanium, nickel ‘solute’ without consideration of sodium segregation. Previous study has proven that it is a more complex system, wherein more than one ‘solute’ metal-ion replaces one ‘solvent’ metal-ion. Manthirm et al. found that when high-voltage cathode LiMn_{1.5}Ni_{0.5}O₄ is simultaneously substituted by Fe and Ga, only Ga-ions preferred to significantly segregate in the surface rather than Fe-ions, meanwhile Fe and Ga-ions both presented a remarkable segregation in case of separated doping by Fe or Ga (Chem. Mater., 2012, 24, 3720-3731). It is documented that when an oxide compound is simultaneously substituted by several other metal-ions, one of them would be finally segregated in the particle surface. Similar to that, titanium ions instead of nickel ions consequently showed a preferential segregation in the surface of NMTN samples. Detailed reasons are still unclear, and will be further investigated via the

combination of experiments and simulations in the future.

The description is included accordingly in the revised manuscript:

Surface segregation is common in oxide compounds and metallic alloys, and the driving force of segregation is the surface energy and the strain energy in the bulk, respectively.^{31,32} A model calculation on surface segregation for oxide solid solution has been demonstrated by Kung in 1981,³³ and it has been concluded that the substantial mismatch between the ionic radii of ‘solvent’ metal-ions and ‘solute’ metal-ions contributes to the surface segregation. In the simple statistical-mechanical mode of segregation,³³ the total free energy F for a system consisting of several elements is written as:

$$F = \sum_i n_i^b g_i^b + n_i^s g_i^s - k_B T \ln \Omega \quad (1)$$

where n_i^b and n_i^s indicate the number of bulk and surface atoms of type i with individual free energies g_i^b and g_i^s , respectively. Ω represents the entropy due to the mixing of the compounds. For a two-component system, an Arrhenius expression is obtained:

$$n_1^s/n_2^s = n_1/n_2 \exp(-H_{seg}/k_B T) \quad (2)$$

where H_{seg} is a heat of segregation. Equation 1 implies that surface segregation is a competition to minimize the total free energy by a maximization of entropy by mixing evenly several elements and a minimization of free energy of each element. From Eq. 2, the enthalpy can be determined as a function of the atomic fraction.³² Mackrodt et al. also measured a heat of segregation for various binary oxide systems, and a very large driving force toward the equilibrium segregation of calcium is demonstrated, namely $-50.7 \text{ kJ mol}^{-1}$ for the heat of Ca segregation in surface of MgO.³⁴ That means if there is little significant segregation, the terminal layer is a polar surface, i.e., a surface is likely to be unstable toward surface reconstruction. These differences at the surface suggest that, under equilibrium conditions, the surface will not have a composition and/or structure representative of the bulk, and the ‘solute’ ions are inclined to segregate in the surface.³² In NMTN samples, ‘solute’ titanium-ions show significant surface segregation despite of the presence of the mismatch between nickel and manganese-ions. This is similar to the previous report, where in case of bidoping Fe and Ga in high-voltage cathode $\text{LiMn}_{1.5}\text{Ni}_{0.5}\text{O}_4$, a preferential segregation of Ga rather than Fe was observed.³⁵ Detailed reasons why only one of doped metal-ions shows a significant segregation will be further investigated in the future. It is well known that surface segregation has a profound impact on the electronic and chemical properties of as-prepared materials, which thus is utilized for desirable materials.³⁶

Question 3: “Furthermore, Mn was partly replaced with Ni and Ti at the same time. Are all the benefits own to the Ti-concentrated interface or partially due to Ni doping? In the manuscript, the functions of Ni and Ti are not distinguished. The contribution of Ni doping is not mentioned. Therefore, it is difficult to say that this manuscript is well organized.”

Reply:

Thanks for your valuable comments. Aiming at addressing the interface concerns associated with the structural degradation by water inserting and manganese dissolution by electrolyte corrosion, a robust and stable interface in electrolyte and moisture air is of great significance for SIBs electrodes, and Ti-concentrated surface arising from surface segregation exactly overcome the above limitations rather than Ni doping. Through series of aging experiment, and electrochemical and thermal tests, we can clearly see that NMTN samples with Ti-concentrated surface can strongly stand up to the corrosion of different environments (air or electrolyte) as well as performing improved thermal stability in contrast to the susceptible surface of NM samples. It should be emphasized that Ni simultaneous substitution makes no difference to the intrinsic interface issues of layered manganese oxides. Ni doping is utilized to balance the charge when manganese-ions are substituted by titanium-ions that mainly present higher valence. Ni simultaneous substitution maintains the high activity of NMTN samples and reduce the Mn(III) distortion in layered bulk. To well distinguish the role of nickel from titanium, we have modified accordingly the manuscript as follows:

In addition to the titanium-segregated interface that greatly suppresses the electrochemical degradation, Ni simultaneous substitution also maintains the charge balance and the high activity of NMTN samples and reduces the Mn^{3+} distortion in layered bulk, both of which jointly contribute to the improved electrochemistry.

The dramatically enhanced cyclic stability further proves the merits of titanium segregation in the surface and Ni substitution in the bulk for NMTN samples.

As mentioned above, a large number of researches refer to the doping in the bulk of layered Na_xMnO_2 , however, less noticed is the fact that the interface concern always inflicts the atmospheric and electrochemical corrosion on layered Na_xMnO_2 . Herein, a Ti-segregated spinel-like surface is ingeniously designed to not only break the bottleneck, i.e., interface issues of layered Na_xMnO_2 , but also demonstrate the best cycling performance among all layered manganese-based cathodes. Therefore, we think this manuscript is compact and carefully constructed, demonstrating the effects of ingenious atomic-scale material engineering and deserving the more attention in the battery community.

In summary, we believe this work becomes much more thorough after adding both of the supplementary data and revised description, we hope the referee would also feel more impressive based on our further improvement and think the work is suitable for publication in Nature Communications.

Referee 3

The authors have answered the first and second comments satisfactorily. However, the referee does not agree with the reply for the third comment. Indeed, $\text{NaMn}_0.8\text{Ti}_0.1\text{Ni}_0.1$ exhibits outstanding electrochemical performance. However, the comparison between NaMnO_2 and $\text{NaMn}_0.8\text{Ti}_0.1\text{Ni}_0.1$ is not suitable. The differences between these two materials are multiple: i) Ti surface segregation, ii) synthesis temperature, iii) crystal structure and iv) chemical composition. In this case, it is quite difficult or even impossible to distinguish the contributions of these differences for the improvement of electrochemical performance.

Reply:

Thanks for your valuable comments. It is true that several differences present in the NMTN and NM cathodes, however, we should emphasize that Ti surface segregation is the determining factor for the improved performance of NMTN. That is because that the common concern of layered manganese oxides is the fragile surface and the related side reaction under moist air and organic electrolyte. Therefore, to construct an environmentally-stable interface against water insertion and manganese dissolution occurring at surface, is of great significance. As we know, the synthetic temperature, $\text{O}3$ or $\text{P}2$ structure, and chemical composition are nearly independent of the surface reactions associated with water insertion and Mn dissolution for layered Mn-based cathodes. Focusing on the influence of interface on electrode performance, we believe that Ti surface segregation could be remarkably distinguished from other factors of NMTN and NM cathodes and dominates the contribution to improved structural stability and electrochemical performance.

Authors added a sentence of “Ni simultaneous substitution also maintains the charge balance and the high activity of NMTN samples and reduces the Mn^{3+} distortion in the layered bulk” in the revised manuscript. However, only one sentence is insufficient to fully compare NaMnO_2 and $\text{NaMnTi}_0.1\text{Ni}_0.1$. Therefore, referee judge that the present manuscript is not suitable for publication in Nature Communications.

Reply:

Thanks for your helpful comments. The common concern of layered Mn(III)-based oxides is the surface side reaction associated with the environmental corrosion. Ni mainly distributes in the bulk rather than the surface, hardly affecting the surface properties of NMTN samples. Ti surface segregation effectively protects NMTN samples against the water insertion and manganese dissolution occurring at the surface. To better distinguish Ti surface segregation from other factors for comparing NMTN and NM cathodes, we accordingly include the description in the revised manuscript:

It is worth noting that besides Ti surface segregation, other differences such as synthetic temperature and chemical composition also present in the NMTN and NM cathodes, however, which are nearly independent of the environmental corrosion occurring at surface. Ti surface segregation dominates the protection against the surface side reactions and potentially contributes to the improved electrochemical performance.

Our findings not only provide a SIB cathode with excellent battery performance, but also represent a major advance in the fundamental understanding of Na-ion electrochemistry. Therefore, we believe that this paper is suitable for publication in Nature Communications.

Referee 3

The revised one is acceptable for publication.

Reply:

Thanks a lot for your comments.